# Functionalized Nanomembranes and Plasma Technologies for Produced Water Treatment: A Review

**DOI:** 10.3390/polym14091785

**Published:** 2022-04-27

**Authors:** Anton Manakhov, Maxim Orlov, Vyacheslav Grokhovsky, Fahd I. AlGhunaimi, Subhash Ayirala

**Affiliations:** 1Aramco Innovations LLC, Aramco Research Center, 119234 Moscow, Russia; maxim.orlov@aramcoinnovations.com (M.O.); vyacheslav.grokhovsky@aramcoinnovations.com (V.G.); 2EXPEC Advanced Research Center, Saudi Aramco, Dhahran 31311, Saudi Arabia; fahad.ghunaimi@aramco.com (F.I.A.); nethajisubhash.ayirala@aramco.com (S.A.)

**Keywords:** plasma, produced water, membranes, desalination, gliding arc, nanofiltration, nanofibers

## Abstract

The treatment of produced water, associated with oil & gas production, is envisioned to gain more significant attention in the coming years due to increasing energy demand and growing interests to promote sustainable developments. This review presents innovative practical solutions for oil/water separation, desalination, and purification of polluted water sources using a combination of porous membranes and plasma treatment technologies. Both these technologies can be used to treat produced water separately, but their combination results in a significant synergistic impact. The membranes functionalized by plasma show a remarkable increase in their efficiency characterized by enhanced oil rejection capability and reusability, while plasma treatment of water combined with membranes and/or adsorbents could be used to soften water and achieve high purity.

## 1. Introduction

The sustainable development of modern society is increasing the energy demand, leading to a higher power being produced [1]. At the same time, the importance of all environmental concerns is growing substantially every year [2]. While attention to renewable sources is great for multiple reasons, the improvements and optimization of the technologies in the current format of the energy sector are also highly essential [3]. This necessitates the need to revise processes in the oil & gas industry towards more sustainable strategies in terms of energy efficiency, process-water reusability, and recycling of chemicals [4].

The water resources have a significant impact on the efficiency of the oil & gas industry, as the energy sector is 6th in the most water-intensive industries with 52,000,000,000 m^3^ of freshwater consumed annually for global energy production [5]. The reuse of wastewater streams became an important research topic with more than 130 000 publications in this field. The number of publications focused on the produced water treatment is also rapidly growing and will be even more significant in the coming years (Figure 1a).

In the oil & gas sector, the produced water is the most significant waste stream, especially for the wells with high maturity [6]. Production activities generate increasing loads of produced water up to thousands of tons each day or more, depending on the geographic area, formation depth, oil production techniques, and age of oil supply wells. The regulations related to the discharge of the produced water are stringent, although they might vary for different countries. For example, in Denmark, it must be below 30 ppm [7]. The Regional Organization for Protection of the Marine Environment (ROPME) issued several regional agreements controlling drilling operations in Arabian Gulf. According to the ROPME protocol concerning Marine Pollution resulting from Exploration and Exploitation of the Continental Shelf, the effluent discharge should contain less than 15 ppm of oil, and and no drilling waste above 40 ppm [8]. The water is produced from different oil fields containing different chemical compositions. Currently, produced water is also known as industrial wastewater containing heavy metals that are toxic to humans and the environment, requiring special processing so that they can be properly disposed [9]. The ever-evolving and increasingly stringent regulatory standards for discharging produced water pose major environmental and economic implications [6]. Hence, the treatment of produced water creates a significant challenge that new materials and technologies must be capable of solving.

The importance of this topic is also supported by the decisive engagement/increase of R&D funds, among international organizations and private companies, with a total number of awarded research grants of about 328 (determined according to the literature analysis using Scopus database). For example, the efforts to enhance the economic efficiency of sustainable produced water treatment processes were supported by different initiatives, including the U.S. Department of Energy (DOE). These initiatives actually aim to develop advanced technologies that can improve oil and gas production while protecting water resources, reducing water use [10].

The topic of produced water treatment was reviewed several times in the last decade, e.g., in [11,12,13]. The water treatment methods are generally based on chemicals, thermal processes, electrodialysis, membrane distillation, etc. However, chemical water treatment and thermal treatment have certain limitations, including economic efficiency and environmental impact (production of chemical, energy-intensive processes). In contrast, membrane technologies, plasma treatment, or combined technologies have excellent potential for oil/water separation, desalination, and heavy metals/toxic chemical removal. The use of plasma technologies for wastewater treatment also became rapidly growing topic with several hundred publications per annum, as shown in Figure 1b.

This review is focused on hybrid strategies and technologies based on nanomaterials and plasma processes with a significant focus on the sustainability of each proposed methodology. Indeed, as the plasma process is an environmentally friendly technology enabling new materials, it has excellent potential to enhance the water/oil separation process, induce new reactive cites to improve the desalination, and, finally, destroy toxic organic chemicals or convert heavy metals into sediments.

## 2. Plasma Modified Membranes Used for Oil/Water Separation

Plasma modification of different materials enables enhanced properties by adjusting surface chemistry and roughness without damaging the bulk properties [14]. The nonthermal plasma is generally used to modify the surface of materials as it will not damage the surface polymeric materials, including temperature-sensitive materials [15]. Nonthermal plasma can be ignited at different pressures by applying the high voltage to the electrodes separated by the gap (filled by noble gases, air or a mixture of precursors) [16]. To obtain homogenous plasma at atmospheric pressure, the surface of the electrodes could be covered by a dielectric layer (to prevent arcing). At the same time, the frequency of the applied voltage can be maintained in the range from a few kHz up to several MHz. It was often used to increase the superhydrophobicity or superhydrophilicity of nanomaterials to develop highly efficient nanofiltration membranes [17,18].

Nanofiltration (NF) is a perspective technology that provides a combined solution to reject both organic and inorganic pollutants, thereby making it more beneficial than ultrafiltration (UF). However, compared to reverse osmosis (RO), the salt and metal ion rejection percentages are typically lower (50–80% vs. 99%). The rejection of organic molecules is also somewhat lower. In comparison, ultrafiltration membranes allow passing almost all components with MW < 10,000 g/mol, while the pore size of nanofiltration membrane is much smaller, allowing many smaller organic molecules to be rejected. As a result, nanofiltration can be classified between reverse osmosis and ultrafiltration on the filtration spectrum [19].

Nanofiltration has a strong potential for oil/water separation, as it allows to obtain cleaner water from oily solutions, including the possibility for in-house recycling [20]. If the discharge of the produced water is preferred, nanofiltration permeate can also be clean enough to meet the stringent discharge requirements [21]. However, different challenges, including economic viability, robustness, and modest efficiency, hinder the large-scale nanofiltration application. This section summarizes the most advanced eco-friendly oil/water separation solutions based on the modified nanofiltration membranes.

A novel environment-friendly method to prepare nanofiltration membranes for highly efficient separation of water-in-oil emulsions was reported by Yang et al. [22], where durable biphilic (oleophilic and hydrophilic) surface was prepared by atmospheric air plasma treatment of cellulose filter paper followed by various grafting procedures. The hydrophobic surface with a microscale hierarchical structure was constructed using long-chain alkyl silane grafting. The authors utilized silanization by acting aminopropyltrimethoxy silane (APTMS), glycidyl propyl trimethoxy silane (GPTMS) and hexadecyl trimethoxy silane (HDTMS) onto plasma-activated SiO_2_ nanoparticles using surface chemistry approach as shown in Figure 2. The samples exhibited excellent separation performance for surfactant-stabilized water-in-oil emulsions with the filtration efficiency of up to 99% and the oil recovery purity > 99.98 wt%. As shown in Figure 2, the water-amine attraction (for APTMS functionalized layers) and oil-alkyl attraction (for the GPTMS side) synergistically destabilize the emulsion and separate the two phases. The formatted coating was very durable against ultrasonication in acetone and can be stored for 12 months without degradation upon aging. The coating also exhibited excellent recyclability (over 20 cycles), although the multi-step process is too complicated and not cost-effective for large-scale application.

Various materials, including fluoropolymers, treated by plasma exhibited enhanced oil-water separation properties. For example, the traditional polytetrafluoroethylene (PTFE) membrane was pretreated using the atmospheric pressure glow discharge (APGD) plasma. Then, the surface of the plasma-treated PTFE membrane was modified by the photo-induced graft polymerization of 2-acrylamido-2-methyl-propyl-sulfoacid (AMPS). The obtained surface demonstrated a highly negative charge, ensuring substantial oleophobic property of the PTFE membrane and leading to improved anti-fouling performance compared to the unmodified sample [23].

A superhydrophilic PVDF membrane obtained by acrylic acid plasma polymerization followed by TiO_2_ nanoparticles self-assembly was employed for oily produced water treatment [24]. The TiO_2_ nanoparticles were immobilized onto the membrane surface via the coordination of Ti^4+^ with a carboxylic group from plasma layers. It is worth noting that the immobilization step did not affect the valence state of Ti^4+^ (Figure 3). The TiO_2_-decorated PVDF membranes exhibited high uniformity and dramatically improved surface hydrophilicity. After modification, the permeation flux was increased more than four times, and the oil rejection rate was higher than 92%.

Lin and co-workers developed Janus membrane composed of the PVDF substrate functionalized with a special silicification layer inducing the asymmetric wetting selectivity [25]. The hydrophobicity of the membrane was facilitated by grafting the hydrophobic perfluorodecyltriethoxysilane, while unilateral plasma etching treatment of modified membrane allowed to induce the hydrophilicity. By tuning the surface chemistry of hierarchical nanosphere-like architecture of a robust membrane with superhydrophobicity/superoleophilicity (perfluorodecyltriethoxysilane modified side), and underwater superoleophobicity/superhydrophilicity (plasma-modified side) were obtained. The membranes showed an extremely opposing surface wettability with a high water/underwater oil contact angle (CA) difference up to 150° due to the exceptionally asymmetric wetting selectivity incorporated with the 15 nm ultrathin silicification layer. As a result, the water-oil separation efficiencies in surfactant-stabilized oil-in-water and water-in-oil emulsions reached 99.8% and 99.0%, respectively. In addition, the authors demonstrated nearly 100% recovery ratio of permeate flux after several cycles of oil-in-water and water-in-oil emulsion filtration tests.

It is worth noting that the multi-step functionalization of PVDF membranes has a particular drawback: this process requires several reactors and a relatively long treatment time. As a result, it increases the production cost. Therefore, many research works were focused on utilizing cheaper materials and improving the cost-efficiency of the modification to one-step processes.

Complimentary to the PVDF, polyethersulfone (PES) ultra- and nanofiltration membranes are attractive materials due to their high efficiency and stability [26]. Sadeghi and co-workers prepared efficient PES ultrafiltration membranes using a corona plasma treatment to reduce membrane fouling during the separation of oil/water emulsions [21]. These PES membranes were fabricated by the phase inversion method and thoroughly analyzed. The authors investigated the effects of solvent in casting solution and corona treatment conditions on membrane surface properties, morphology, and separation performance. The corona treatment induced the grafting of hydrophilic OH groups leading to the higher water flux in all configurations of membranes without significant changes in oil rejection. Finally, the rejection rate up to 99.5% was observed at the oil/water permeability range from 34 to 70 (L m^−2^ h^−1^ bar^−1^). Similar approach was exploited by Adib et al. [27] to explore the potential of corona plasma-treated PES membranes for water-oil separation. By treating the membrane in corona discharge at 360 W for 6 min, the authors achieved their most promising sample with the rejection rate of 92%. Membranes exhibited mean pore size~30 nm at a porosity of 85%. To enhance the grafting of hydrophilic COOH groups and, additionally, to utilize CO_2_ to improve the sustainability of the approach further, carbon dioxide can be admixed to the plasma-forming gas mixture. CO_2_ plasmas were used to modify hydrophobic polysulfone ultrafiltration membranes to create hydrophilic surfaces throughout the membrane structure [28]. The water contact angle (WCA) of the upstream side of the membrane (facing the plasma) decreased to zero after treatment and remained at the same condition even after several months of aging.

Fedotova and co-workers employed radio-frequency (RF) glow discharge to treat polysulfonamide membranes (pore size of 0.01 μm) [29]. Low-pressure RF discharge in an argon and nitrogen atmosphere at an anode voltage of U_a_ = 1.5 kV and contact time τ = 1.5 min led to a decrease in the contact angle from 59.6° to 47.9°, and grafting of oxygen groups. Experiments on the membrane separation of a 3% oil-in-water emulsion have been conducted. The results have shown that the use of plasma-treated polysulfonamide membranes leads to the intensification of the oil/water separation process.

In order to enhance the wettability or superhydrophobicity, many researchers prefer to use the materials with advanced nanoporous structures, e.g., electrospun nanofibers. Indeed, the production of nanofibrous membranes has approached the commercial scale processes with many products, including air-filters and masks, showing high cost efficiency [30,31,32]. Thus, the implementation of nanofibrous membranes for water treatment offered significant potential.

Nanofibrous membranes with superhydrophilic/superoleophobic surfaces were constructed by grafting acrylic acid onto low-pressure nitrogen plasma treated electrospun polystyrene/polyacrylonitrile (PS/PAN) nanofibrous foils as shown in Figure 4 [33]. The WCAs of the PS/PAN membranes decreased to 0° after grafting treatment of acrylic acid, thereby proving that the modification improved the surface hydrophilicity of the membranes due to the introduction of hydrophilic groups. Oil/water permeation tests confirmed a high oil/water separation potential of plasma-treated PS/PAN membranes. The results showed that these membranes effectively separated the layered oil/water mixture with permeate flux up to 57,509 L m^−2^ h^−1^, while high fluxes of 1390–6460 L m^−2^ h^−1^ were observed for the separation of different oil-in-water emulsions. The prepared membranes exhibited superhydrophilic and underwater superoleophobic surfaces, which could prevent oil droplets from adhering to the surface, thereby contributing to membrane anti-fouling. In the separation step, the resultant membranes, solely driven by gravity, had high separation efficiency as well as one to two orders of magnitude higher fluxes than that of traditional polymeric filtration membranes with similar permeation properties [33].

Notably, the membranes maintained high flux and efficiency even after several separation cycles to demonstrate excellent recyclability. Therefore, such flexibility in reusing these membranes is expected to provide potential cost-effective materials for oil/water treatment.

The strategy shown above for PS/PAN membrane modification by acrylic acid grafting using plasma activation required a wet chemical step that is preferably switched to a gas-phase process or a single-step plasma modification for better cost-efficiency. Of course, the one-step atmospheric plasma method would be the most desirable, but it is pretty challenging at the same time [34,35]. Moreover, the wet chemical steps are often eliminated by low or atmospheric pressure plasmas to improve the economic viability.

Several researchers demonstrated the capability of atmospheric plasma for modification of membranes to stimulate the oil/water separation. Chen et al. proposed the environment-friendly oil/water separation methodology based on atmospheric plasma-treated nylon meshes [36]. Nylon fibers were exposed to atmospheric pressure plasma for surface modification, leading to micro/nanostructures and oxygen-containing groups. Consequently, the functionalized meshes possess superhydrophilicity in air and thus superoleophobicity underwater. The obtained materials exhibited an efficiency above 97.5% for various oil/water mixtures. Moreover, the authors showed that the functionalized nylon meshes have excellent recyclability and durability in oil/water separation. Fedotova et al. optimized the plasma modification of polysulfonamide membranes (with a pore size of 0.01 μm) for the separation of model “oil-in-water” emulsions containing 3% of industrial oil [37]. The atmospheric pressure plasma torch was ignited for short time periods from 1 to 7 min at the anode voltage ranging from 1.5 to 7.5 kV. The highest performance was observed in the case of membranes treated with plasma for 7 min, leading to increased membrane efficiency from 90 to 99% owing to the surface hydrophilization. Indeed, the membrane exhibited a water contact angle 59.6° before treatment, and after a 4-min plasma treatment at 7.5 kV it decreased to 19.5°. The authors also confirmed that the plasma treatment leads to a change in the surface structure of the membranes, namely, a decrease in their roughness. The membrane internal microstructure also undergoes changes that increase their crystallinity.

You et al. employed He/CH_4_/C_4_F_8_ atmospheric plasma deposition to modify the surface of stainless steel meshes [38] and applied them for oil/water separation. The fine-tuning of the plasma process parameters enabled the selective functionalization of each side of the membranes, leading to superhydrophobic-superoleophilic and superhydrophobic-oleophobic sides. The authors demonstrated the successful separation in a simple test, where a 50 mL oil-water solution in 6 min was treated. According to the FTIR analysis, approximately 88% of the hydrocarbons in the oil-water mixture were separated in 6 min without any external forces, and no water was detected in the collected oil. The separation efficiency of up to 99% has been achieved. A similar concept of oil/water separation using oxygen plasma treatment of Cu coated meshes to induce hydrophobic and oleophilic behaviors is presented by Agarwal et al. [39]. This concept’s advantage is the possibility of using cheaper and more robust steel grades for separating meshes.

Sometimes, produced water may contain special chemicals, e.g., hydrolyzed polyacrylamide (HPAM), which is added to the displacing fluid to enhance the mobility and extraction of the oil phase. Optimized ultrafiltration, using ceramic membranes with a surface pore size of 15 kDa, to effectively separate HPAM from produced water was demonstrated by Ricceri et al. [40]. The precipitation and ultrafiltration may be used in sequence as they complement each other in several ways. Moreover, ultrafiltration membranes with 15 kDa cut-off provided the best combination of productivity and HPAM removal (92%). However, the authors highlighted the problem of membrane fouling under severe conditions.

## 3. Nanofiltration Membranes for Desalination

The rejection of salts from the produced water is very challenging task. However, a sustainable nanofiltration approach using modified nanomaterials significantly progressed to solve this problem by using various advanced 2D nanomaterials, including graphene and boron nitride [41,42]. An exciting approach based on membrane crystallization was demonstrated by Ali et al. [43]. The authors employed membrane crystallization for desalination and salt recovery from produced water streams at a semi-pilot scale using convetional polypropylene or in-lab prepared PVDF hollow fiber membranes. The experiments were carried out in lab scale and semi-pilot scale. It is worth noting that the salts may also be in high demand, and the recovery of ions can be highly interesting if economically viable. Unlike conventional crystallizers, in membrane crystallization, well-controlled nucleation and crystal growth are achieved through a uniform evaporation rate through the membrane pores. The crystals recovered by membrane crystallization have higher purity and narrow size distribution. The recovered crystals were sodium chloride with high purity (>99.9%). It was demonstrated experimentally that at a recovery factor of 37%, 16.4 kg NaCl per cubic meter of produced water could be recovered.

Another method to extract salts from produced water is the airgap membrane distillation. Woo et al. modified the PVDF membrane by using electrospinning of PVDF, modified the surface of the nanofibrous foil, and evaluated the membrane’s performance using real reverse osmosis brine with a produced water as a feed [44]. The results suggest the successful chemical modification of the membrane by plasma treatment without significantly altering the morphology and its physical properties, providing increased fluorination primarily through the formation of CF_3_ and CF_2_-CF_2_ bonds, with the treatment duration. The optimal 15 min plasma treatment induced the omniphobic property of the membrane with the highest hydrophobicity and high wetting resistance to low surface tension liquids such as methanol, mineral oil, ethylene glycol, and consequently high liquid entry pressure (187 kPa). These improved properties translate to high flux (15.28 L/m^2^h) and salt rejection (~100%) performances of membrane even with the addition of up to 0.7 mM sodium dodecyl sulfate in the reverse osmosis brine feed during air gap membrane distillation.

## 4. Plasma-Aided Technologies for Produced Water Treatment

The plasma-aided water treatment is generally achieved by activating highly efficient adsorbents or direct plasma treatment of water by atmospheric discharges. Each technique has its own advantages and drawbacks, and the most noticeable results are reviewed below.

### 4.1. Plasma-Activated Adsorbents

The economic aspects for the oil/water separation play a crucial role in the evolution of produced water treatment technology. Often cost-effective reusable materials with lower performance may have gained economic benefits against polymeric membranes based on PVDF, fluoropolymers, polysulfone, or other expensive materials. Hence cheaper feedstock material can play a crucial role in reducing costs and enhancing the business value for membrane oil/water separation. In this regard, the two-step surface modification of polyamide meshes and nonwoven fabrics for oil/water separation investigated by Zhao et al. attracted extreme attention from the scientific community [45]. Their methodology was based on pre-etching the polyamide surface using plasma treatment and coating of a pre-etched surface by eco-friendly polydopamine (PDA)/cellulose, as shown in Figure 5. The pre-etching increased the surface roughness, which further improved the underwater superoleophobicity of the coating. Therefore, the modified polyamide separated various oil/water mixtures and showed a higher intrusion pressure than the original sample and the samples, which were only etched or coated. The grooves on the surface that resulted from the pre-etching prevented the coating from peeling off. In durability tests, after six repeated uses, the modified nonwoven sample lost its underwater oleophobicity due to severe oil fouling, coming to a complete failure in oil/water separation. After 19 cycles, the modified mesh could still separate a certain amount of oil/water but showed reduced intrusion pressure because of slight oil contamination. Filters with different structures, like meshes with one layer of pores and nonwoven fabrics with complex three-dimensional pores, had different oil fouling levels that affected oil/water separation. The recoverability of filters from oil contamination should be considered for practical applications.

Anupriyanka et al. presented the efficient and cost-effective sustainable oil/water separation by fabricating innovative oil-recovery materials (fluorine and silane-free) using an environment-friendly route [46]. Researchers produced superhydrophobic PET (polyethylene terephthalate) fabric by a DC glow discharge oxygen plasma treatment followed by the incorporation of the superhydrophobic agents. The WCA of this fabric was 163°. The prepared fabric possesses excellent repellency to water while absorbing oil and hydrocarbons. Indeed, PET is a significantly cheaper material, and even disposed fabrics can be used as a feedstock to fabricate such adsorbents.

A superhydrophobic cotton nonwoven fabric aimed at oil-water separation was prepared by a two-step strategy based on (1) atmospheric-pressure N_2_/O_2_ plasma treatment and (2) graft polymerization of siloxane [47]. Different process conditions (plasma power, composition of the plasma gas) influenced the contact angle, stability, surface morphology of the hydrophobic coating, and the growth of silica nanoparticles. The water contact angle of treated nonwoven was up to 155°. The superhydrophobic nonwovens treatment showed excellent stability toward severe acid and alkaline conditions and subsequently the resulting fabrics may be used under harsh environmental conditions. The modified cotton nonwoven exhibited an oil/water separation efficiency higher than 97%, and the result was repeated at least ten times for each sample.

### 4.2. Direct Plasma Treatment of Water

Atmospheric pressure discharges can directly decompose the organic pollutants. The decomposition of model organic pollutants: chloroform, benzene, and methanol through the gliding arc (GA) plasma technology was reviewed by Gong et al. [48]. The conventional GA reactor is depicted in Figure 6. It is equipped with an AC power supply and flat electrodes contributing to the dispersion of heat and restriction of the gas flow. When a high voltage is applied, the gas flow initiates the GA plasma.

GA plasma devices can be directly used to treat the organic waste gas derived from industrial emissions and be coupled with other technologies or devices, such as catalysts, washing towers, combustion chambers, and activated carbon. GA plasma has extensive adaptability for the types and concentrations of various pollutants, including linear hydrocarbons, aromatic hydrocarbons, halogenated compounds, ethers, and alcohols. Nevertheless, although the gliding arc requires less energy for organic waste gas treatment than other nonthermal technologies, it is still a little higher considering the practical engineering applications. It is worth noting that the plasma treatment can be used to treat produced water and emulsions of water in heavy oils, allowing the separation of oil from water and improving oil properties, as shown elsewhere [49].

Plasma discharges were also used to soften produced water, and the investigation of such a process was funded by US Department of Energy (US DOE) [50]. It was shown that plasma treatment enables the decrease of bicarbonate ions down to 100 ppm.

US DOE has also investigated novel processes for water softening [51]. Specifically, in the plasma discharge project, as mentioned above, the research was focused on reducing the “temporary hardness” (caused by the presence of calcium and magnesium bicarbonate). In this DOE-supported research by Drexel University, plasma arc discharges were used to dissociate and remove bicarbonate ions. In turn, this effect helps prevent fouling from the calcium carbonate scale providing a more prudent strategy for fouling prevention.

Later, Cho et al. investigated stretched arc plasma to increase the volume of produced water treated by plasma, increasing the practical efficiency of the process [52]. Stretching of an arc discharge in produced water was accomplished using a ground electrode and two high-voltage electrodes: one positioned close to the ground electrode, and the other placed farther away from the ground (Figure 7). The contact between the arc and water significantly increased, resulting in twice more efficient removal of bicarbonate ions from produced water when compared to plasma with and without stretching. The usage of plasma treatment for removing bicarbonate ions was further studied by Kim [53]. Although the results were quite promising, the energy cost for this method to treat the produced water was too expensive, and there were no recent publications after 2016.

Choi et al. analyzed the decomposition of water-insoluble 1-decanol by a water plasma system operated at atmospheric pressure [54]. 1-decanol was dispersed in water by a surfactant generating an oil-in-water emulsion. The emulsion was used as the water plasma jet’s feeding liquid and plasma-forming gas (Figure 8). A high decomposition rate of over 99.9999% was achieved by converting 1-decanol emulsion into H_2_, CO, CO_2_, CH_4_, condensed liquid, and solid-state carbon despite relatively low input power (<1 kW). The main organic substance in the treated liquid of 1-decanol emulsion was methanediol produced by the hydration of formaldehyde. The authors concluded that the gas conversion rate of carbon in 1-decanol and the removal rate of total organic carbon concentration were increased by increasing the arc current due to enhanced O radicals in the high temperature of the water plasma jet.

The decomposition of organic pollutants (dyes as a model substance) in water under plasma micro-discharges was studied by Wright [55]. The low-temperature plasma contacting with aqueous solutions, including organic dyes, was analyzed. Using atmospheric plasma for cleaning water was demonstrated by relatively good energy efficiency: 1.3–12 g of removed pollutants per 1 kWh.

Millie et al. and Lemont et al. designed a plasma tool ELIPSE to decompose organic impurities in water using an atmospheric plasma torch [56,57]. The ELIPSE process is a novel technology of organic liquid destruction involving a thermal plasma working under a water column, ensuring the cooling, filtration, and scrubbing of the gases coming from the degradation (Figure 9). The ELIPSE demonstrated the ability to destroy the pure organic liquids and then eliminate the organic compounds remaining in the aqueous solution through plasma’s thermal or radiative properties. Preliminary tests have shown how efficient the process is to destroy the organic liquids when they are directly fed in the plasma. By applying plasma for 60 min, the concentration of organic contaminants decreased from 1200 to 400 ppm.

Yu et al. studied the decomposition of different polycyclic aromatic hydrocarbons (acenaphthene, fluorene, anthracene, and pyrene) by DC gliding arc discharge. The results indicated that the highest destruction rate was achieved with oxygen as a carrier gas and the external resistance of 50 kΩ independently of the type of hydrocarbons. Furthermore, experimental results suggest that the destruction energy efficiency of gliding arc plasma can be improved by treating higher concentration pollutants [58].

The efficient decomposition of toluene in a gliding arc was shown by Du et al. [59]. The toluene removal efficiency increased with the inlet gas temperature, while the presence of water vapors accelerated the toluene decomposition in the plasma. The energy efficiency was 29.46 g per 1 kWh at a relative humidity of 50% and a specific energy input of 0.26 kWh/m^3^, which is higher than other types of nonthermal plasmas. The primary gas-phase decomposition products were determined using FT-IR analysis of the gas components: CO, CO_2_, H_2_O, and NO_2_. Some small deposits of benzaldehyde, benzoic acid, quinine, and nitrophenol were found in the reactor. The authors concluded that these are the minor products from the reaction of toluene with radicals.

## 5. Combined Techniques for Efficient Water Treatment

The increase of the water quality purification related to the final use of the treated water might be extremely challenging to achieve in an economically viable scenario if a single technology is used (e.g., only sorption or filtration). However, the researchers recently implemented an exciting approach based on a combination of various techniques: plasma and sorption of membranes and photocatalysis. These combination approaches allow separating or adsorb parts of the pollutants and decomposing the rest of the contaminants. For example, Gushchin et al. developed a combined process involving the sorption of oil (C_22_H_38_) on a sorbent (diatomite) followed by regeneration of the sorbent by plasma-oxidative destruction in an atmospheric dielectric barrier discharge (DBD) [60]. The reactor depicted in Figure 10 was composed of a glass tube diameter of 22 mm, which served as a dielectric barrier; the outer electrode was made from the aluminum foil and located outside the glass tube, and the internal electrode, a 15 mm diameter aluminum rod. The sorbent was placed inside the reactor along the length of the discharge zone.

The energy efficiency of the decomposition was 0.169 molecules of oil per 100 eV of input energy. It was shown that the complete deterioration of oil on the diatomite surface reached after 5 min DBD treatment, while the decomposition products were water-soluble and non-toxic carboxylic acids, aldehydes, and CO_2_. The complete removal of acids and aldehydes requires a time of about 40 min [60].

Another emerging sustainable approach for decontamination of water is a photocatalytic membrane treatment. Photocatalytic membrane reactor (PMR) is an emerging green technology for removing organic pollutants, photoreduction of heavy metals, photo-inactivation of bacteria, and resource recovery.

The combination of photocatalysis and membrane separation improves the removal of contaminants and alleviates membrane fouling. However, the turbidity and color of produced water reduce the efficient light transmission. Efficient pretreatment improves the resiliency for produced water treatment and minerals recovery using PMR processes [61].

Recently, Butman and co-workers compared photocatalytic, plasma, and combined plasma–photocatalytic water treatment processes using a model pollutant, Rhodamine B dye, solutions with a concentration of 40 mg/L [62]. The TiO_2_-pillared montmorillonite was used as a photocatalyst, while plasma was ignited in dielectric barrier discharge (DBD) plasma. By using the combined DBD and photocatalysis process, a significant increase of degradation efficiency has been observed: combined approach (100%, 8 sec), plasmolysis (94%), and UV photolysis (92%, 100 min of UV irradiation). In contrast to photolysis, destructive processes are more profound and lead to the formation of simple organic compounds such as carboxylic acids. The plasma–catalytic method enhances by 20% the energetic efficiency of the destruction of Rhodamine B compared to simple DBD plasma. The efficiency of dye destruction with the plasma–catalytic method increases with specific surface area and total pore volume, and the size of the TiO_2_ crystallites. This approach may bring significant added value for produced water treatment if the methodology is further optimized.

Implementing the oxidation to the produced water treatment system may provide additional advantages in terms of continuous process reliability. For example, the advanced oxidation processes based on reactions with ozone and Fenton’s reagent efficiently clean the polluted water streams, as revealed by Simões et al. in their bibliometric study [63]. They indicated the increase of publications dedicated to this approach in the last few years, although the first work appeared only in 1995. However, the selected works revealed the limitation of these processes due to the high costs with energy consumption.

Finally, the problem of low water flux attributed to commercial ceramic membranes applied in the treatment of produced water was also studied using complex combined technologies [64]. For minimizing this problem, the titanium dioxide (TiO_2_) nanocomposites, synthesized via a sol-gel method, were deposited on the active layer of the hydrolyzed bentonite membrane [65]. The grafting time of TiO_2_ nanocomposite improved the performance of the coated bentonite membranes. The pure waterpermeability performance showed an increment from 262.3 L h^−1^ m^−2^ bar^−1^ (pristine bentonite membrane) to 337.1 L h^−1^m^−2^bar^−1^ (bentonite membrane with TiO_2_ grafted for 30 min) and 438.3 L h^−1^m^−2^ bar^−1^ (TiO_2_ grafted for 60 min). The oil rejection performance also revealed an increase in the oil rejection performance from 95 to 99%. These findings can be an excellent example to further investigate and exploit the advantages of modified ceramic membranes in produced water treatment.

## 6. Discussion

The produced water treatment solutions based on plasma-aided technologies and nanomembranes show serious potential for industrial applications. However, it is obvious that not every kind of treatment would reach the stage of commercialization. To sum up the advantages and limitations of all of the reviewed technologies, the oil/water separation and desalination technologies are listed in Table 1 and Table 2, respectively.

Proposed technologies can provide the oil/water separation process with high efficiency. The applicability of the proposed methods would greatly depend on the pre-treatment of an effluent and the remaining oil contaminants. In this review, the water/oil separation by most of the membranes allowed to achieve the oil rejection rate ranging from 92% to 99.8%. Hence, knowing the ROPME protocol standard value of 15 ppm (oil in water), it is quite easy to estimate the range of maximum oil concentrations passing membranes. The very well-known formula for oil rejection rate available elsewhere [68] and is presented below:$$Rejection\;Rate=\left(1-\frac{C_{permeate}}{C_{feed}}\right)\times 100\%$$

*C_permeate_* and *C_feed_* are the oil in water concentrations in the permeate and the feed, respectively. Hence, for the technologies exhibiting a rejection rate of 92%, the maximum concentration in the feed can be 187 ppm to ensure the oil concentration in the permeate is below 15 ppm. For the technologies exhibiting a rejection rate above 99%, the *C_feed_* may exceed 1500 ppm. Therefore, the majority of reviewed technologies may satisfy the requirements for the effluent discharge composition according to ROPME protocol.

The implementation of any new water treatment technologies is often depending on the economic, environmental, and technological advantages proposed by the innovative approach. The reviewed technologies are still being tested in a lab or at the semi-pilot scale setups. Hence, the discussion related to the potential economic effects are generally based on a very rough estimations and have low accuracy. Nevertheless, it is worth noting that a great economic potential of some techniques can be predicted based on techno-economic assessments available in the literature. For example the techno-economic effect of the implementation of nanofiltration membranes has been shown by Wenzlick and Siefert [69]. They have modeled the base (without nanofiltration) and advanced (with nanofiltration) processes to model the desalination of produced water. The cost related to NF process was estimated as 0.18 ± 0.07 $/m^3^. The usage of NF process allowed to save the cost of chemicals and turn the economically negative case (−0.98 $/m^3^) to the positive one with a revenue of +0.6 $/m^3^. Hence, the implementation of nanofiltration may add very significant economic advantage to the produced water treatment business case. Furthermore, the costs related to the synthesis of nanofiltration membrane can be further reduced by the means of implementing cheaper recyclable materials as a feedstock for nanomembranes. The techno-economic assessment of using a direct plasma treatment to purify produced water is not available in the literature, most probably due to a very complicated capital costs projection. However, the operational costs can be roughly estimated on the basis of the power consumption estimated for lab-scale setups. Due et al. indicated the power 29.46 g/kWh of organic contaminants is decomposing in their plasma setup [59]. This means that to treat 1 m^3^ of produced water containing 400 ppm of organic contaminants, the required energy will be 400/29.46 = 13.58 kWh. Taking into account energy price at the level 50 to 100 $/MWh, the energy part for operational costs would be 0.68 to 1.36 $/m^3^. The energy consumption for combined plasma-adsorbent technology [60] calculated by a similar procedure would be 21.0 kWh/m^3^. The price for consumed electricity would be in the range from 1.05 to 2.10 $/m^3^ if the same range for electricity pricing is taken into account.

The estimated produced water treatment levelized costs for membrane distillation and evaporation/crystallization processes are 8.1 and 22.1 $/m^3^, respectively [70]. Therefore, the nanofiltration and plasma-based water treatment processes have evident potential to compete with many water treatment technologies in terms of business efficiency.

It is worth noting that presented methodologies would require pre-treatment of the produced water preliminary to the nanofiltration or plasma processing. The variety of the pre-treatment methods, required for the upscaled process cannot be considered at a present technology readiness level as it would be dependent on the properties of the effluent. The pre-treatment of produced water may include de-emulsification, mechanical treatment, or chemical treatments. Generally, the nanomembranes or plasma technologies cannot be considered as a stand-alone one step solution. The produced water treatment should be composed of a combination of several techniques and processes in order to fulfil the requirements for the water quality and cost-efficiency.

## 7. Conclusions & Outlook

This work reviewed multiple techniques for produced water treatment based on various environment-friendly methods and summarized their advantages and drawbacks. It is crucial to summarize the most recent research advancements and suggest the perspectives for further development of this topic. The solutions for produced water treatment would require the proposed technologies to have several essential attributes, such as:(1)Being reliable and cost efficient (the solution should not deteriorate EBITDA)(2)Capable of yielding high separation/recovery efficiency in compliance with the Zero Discharge Liquid (ZLD) approach(3)Provision of a positive impact on the environment (the solution cannot rely on highly toxic chemicals or high CO_2_ emissions due to low energy efficiency)(4)Priority usage of the sustainable materials and technologies employed to realize the proposed solution (the utilization of renewable feedstock or recycled materials are preferable)

It is highly challenging to comply with all of these attributes, and all techniques reviewed in this work need to be optimized. Yet, nanomaterials, surface treatment techniques, and plasma discharges have great potential to become reliable, cost-effective, and environment-friendly solutions for produced water treatment if the following challenges are overcome:(1)Production of the nanomaterials (nanofibers, nanoparticles) used for the membranes or adsorbents must be optimized and upscaled to reduce the current cost. This may happen naturally similar to the witnessed continuous decrease in carbon nanotubes cost.(2)Optimizing the plasma setup to decrease the power consumption and prevent the extreme energy loss for heating. The modeling of efficient plasma setups using computation fluid dynamics and plasma chemistry kinetics must be performed to reach good power efficiency.(3)Combining and upscaling of technologies to accumulate quantitative data for the calculation of energy efficiency and evaluate the economic viability for large scale practical applications.

From these perspectives, it is recommended that multi-step modular approaches based on synergetic effects achieved by membrane distillation, adsorption, and plasma or photocatalytic treatment for regeneration of active materials/membranes have excellent potential. They are likely going to gain increasing attention from researchers and engineers in the coming years.

## Figures and Tables

**Figure 1 polymers-14-01785-f001:**
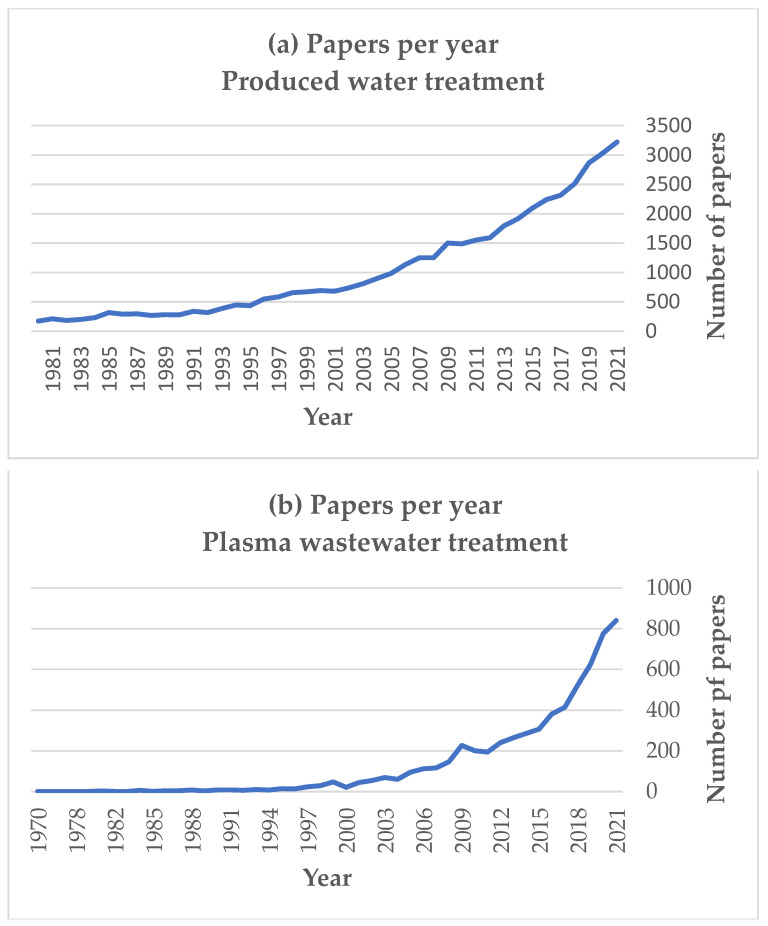
The evolution of publications number (per annum) related to produced water treatment (**a**) and plasma wastewater treatment (**b**). The data was acquired from Scopus database.

**Figure 2 polymers-14-01785-f002:**
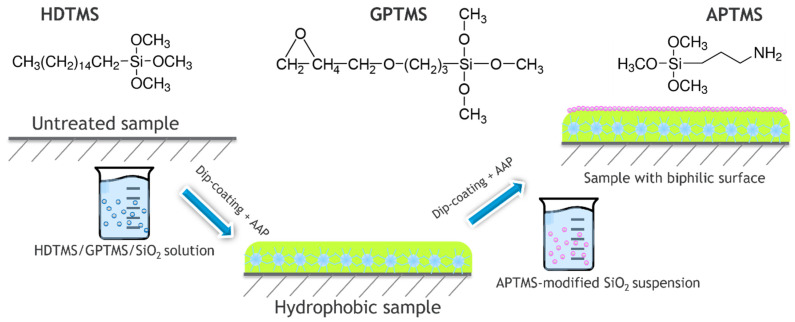
Scheme for the nanofiltration membranes preparation using atmospheric air plasma and modifications by dip-coating.

**Figure 3 polymers-14-01785-f003:**
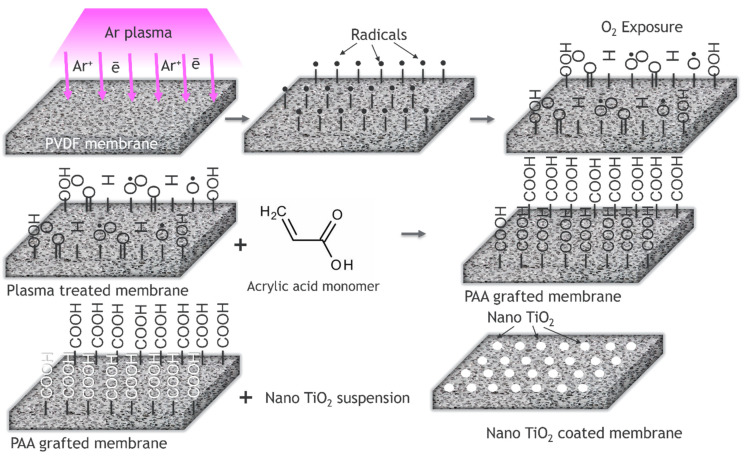
Preparation of nanofiltration membrane by TiO_2_ self-assembling on a PVDF surface.

**Figure 4 polymers-14-01785-f004:**
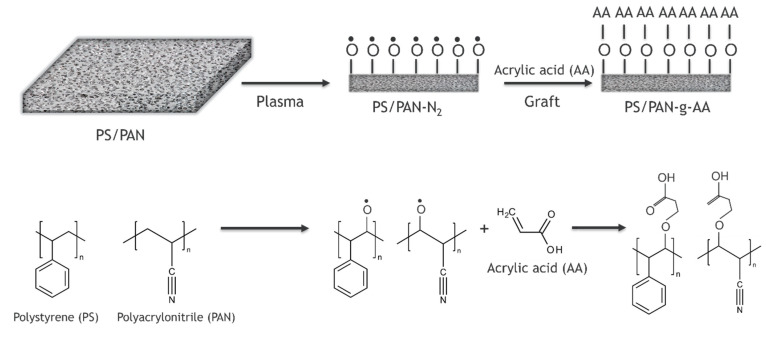
PS/PAN membrane modification by acrylic acid grafting using plasma activation by grafting of radical Oxygen species (O^*^).

**Figure 5 polymers-14-01785-f005:**
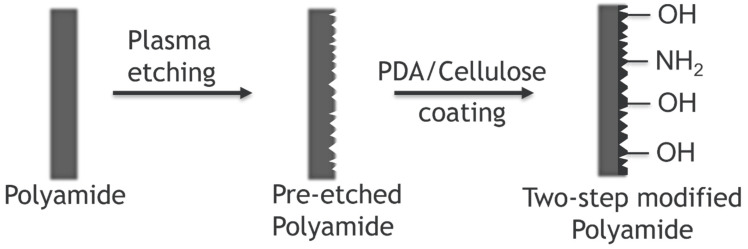
A fabrication of hydrophilic polyamide fabrics for produced water treatment.

**Figure 6 polymers-14-01785-f006:**
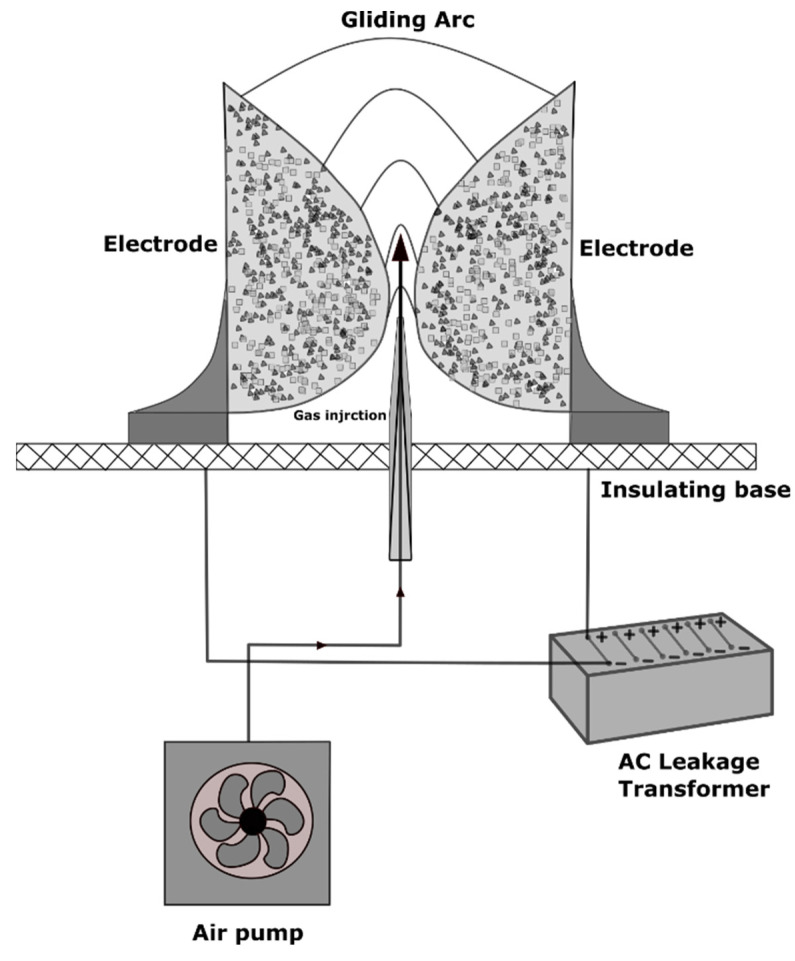
A typical gliding arc setup scheme.

**Figure 7 polymers-14-01785-f007:**
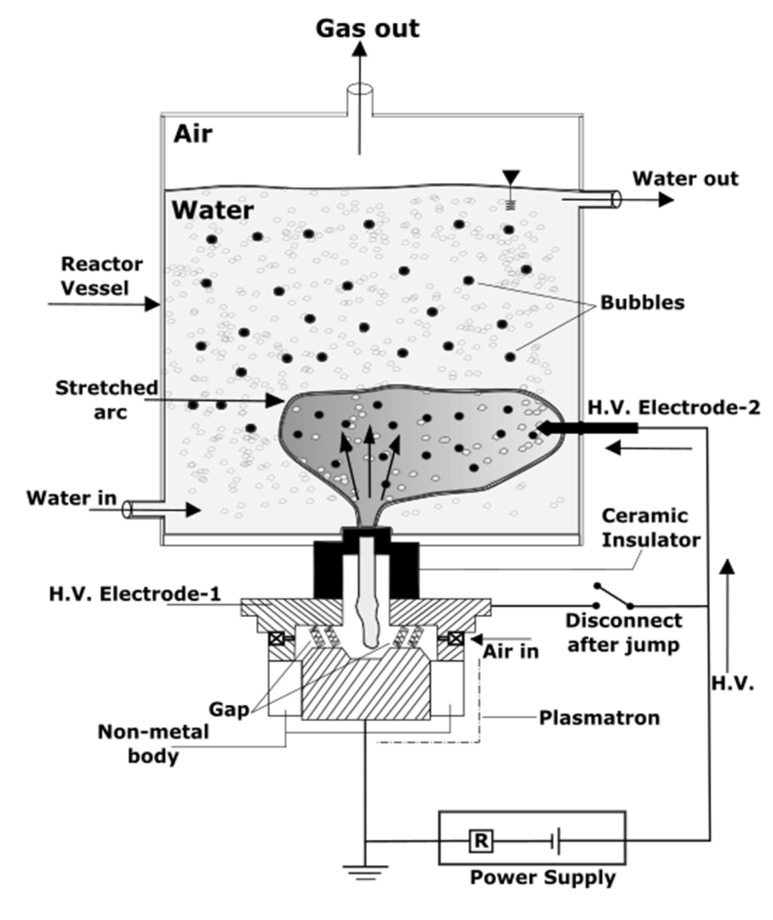
The scheme of a stretch arc plasma applied for produced water treatment.

**Figure 8 polymers-14-01785-f008:**
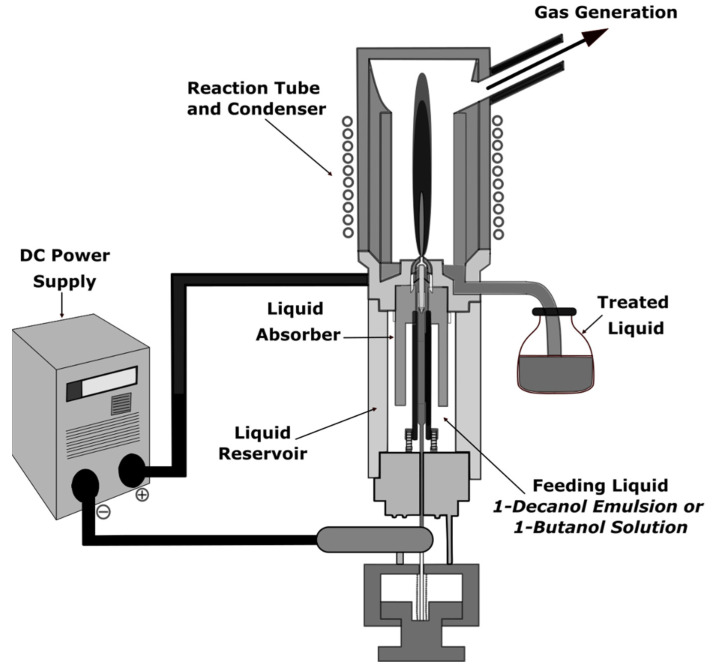
Atmospheric plasma jet setup for water purification.

**Figure 9 polymers-14-01785-f009:**
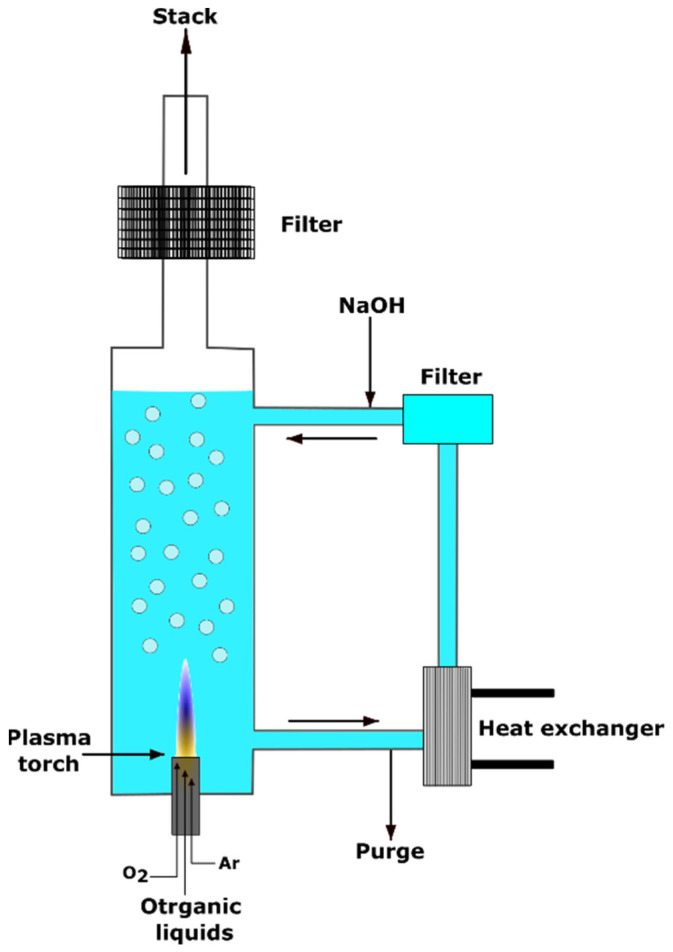
Atmospheric plasma torch employed for the destruction of organic pollutions in water.

**Figure 10 polymers-14-01785-f010:**
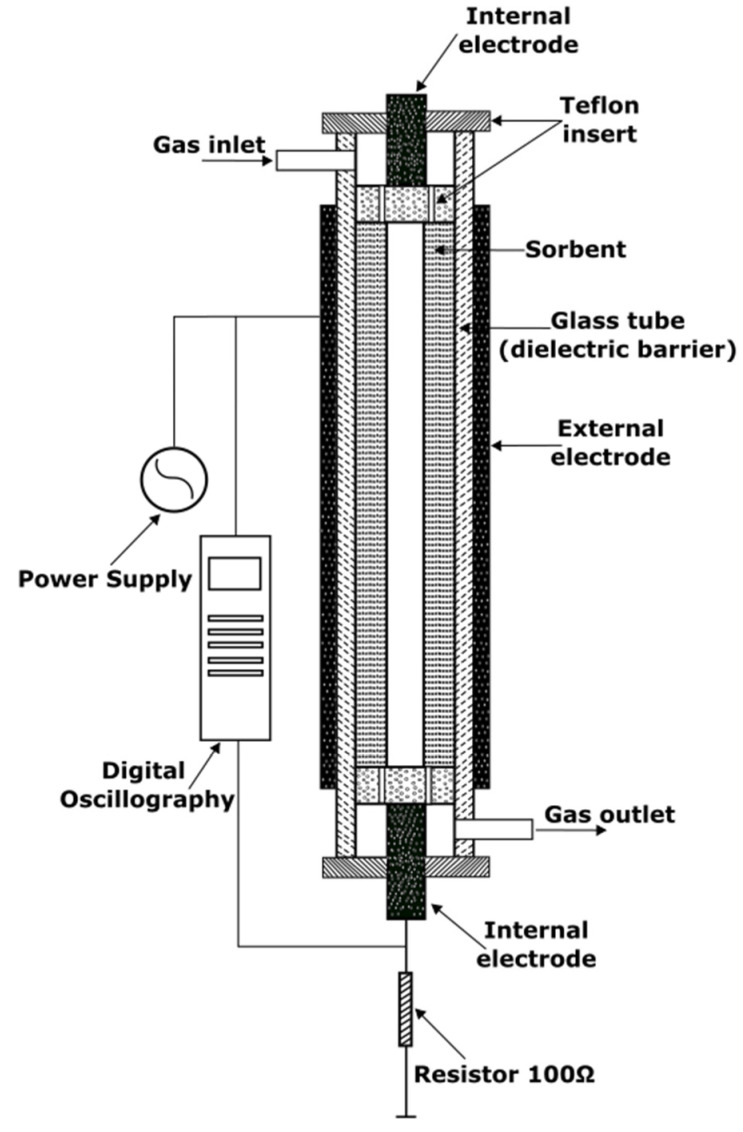
A scheme of combined DBD plasma-sorption technology used for water treatment.

**Table 1 polymers-14-01785-t001:** Summary of the oil/water separation technologies.

Technology	Oil Rejection Rate/Decomposition Rate	Water Flux L·m^−2^·h^−1^	Advantage	Limitation
Superhydrophilic PVDF membrane with TiO_2_ nanoparticles [24]	92%	63,492.Four times higher than pristine membrane	Durability, high permeability.	Multi-step process involving expensive materials.
RF plasma modified polysulfonamide membrane for oil/water emulsion separation [37]	Up to 99%	Up to 30	Simple technology	Low productivity
Superhydrophilic plasma modified PS/PAN nanofibrous membranes for layered oil separation [33]	Up to 99.8%	57,509	Very high oil rejection rate and water permeability	Poor durability: 5–10 cycles only.
Superhydrophilic PS/PAN nanofibrous membranes for water/oil emulsions separation [33]	99.5%	Up to 6460	Very high oil rejection rate and water permeability	Poor durability: 5–10 cycles only.
Superhydrophobic plasma modified steel meshes [38]	Up to 99%	-	Quick separation of water/oil mixtures, simple methodology	No available data for emulsion tests
Superhydrophobic nonwovens	97%		Simple method	The industrial implementation is difficult
GA plasma treatment of produced water [56]	66.7%	-	Decomposing organic chemical contaminations and salts	High remaining concentration of organic contaminations in the discharged effluent, high power consumption.
Combined plasma-adsorbent approach [62]	94%	-	High decomposition rate	Slow process
DC gliding Arc [58]	Up to 98.5%	-	Robustness, high decomposition rate	High power consumption

**Table 2 polymers-14-01785-t002:** Summary of the desalination technologies.

Technology	Salt Rejection Property	Water Flux L·m^−2^·h^−1^	Advantage	Limitation
PVDF hollow fiber membrane [43,66]	~100%	15.28	Straightforward technique, clear pathway for commercialization	Questions about cost efficiency and durability.
Stretched Arc Plasma treatment [52]	Removal of bicarbonate ions as low as100 ppm	-	Robust technique	High power consumption
Distillation by a PVDF nanofibrous mebrane [67]	>99.9%	-	Robust technique with high productivity	Unclear recycling pathway and cost-efficiency, due to a high cost of the PVDF nanofibers

## Data Availability

Not applicable.

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
