# Peer review of "Functionalized Nanomembranes and Plasma Technologies for Produced Water Treatment: A Review"

_polymers, 2022, doi:10.3390/polym14091785_

Round 1
Reviewer 1 Report
This review presents innovative practical solutions for oil/water separation, desalination, and purification of polluted water sources using a combination of porous membranes and plasma treatment technologies. Results of this review may have important application in produced water treatment. Authors may wish to consider the following in revision of their manuscript.
- Please comment on the cost of using the proposed technologies.
- Please list effluent standard of produced water in author’s country.
- Please comment whether the proposed treatment technologies can meet effluent standard.
- Please include a table to summarize treatment performance of using proposed technologies and other commonly used treatment methods reported in the literature.
- Please include pre treatment needed for the proposed treatment method reviewed in this paper.
Author Response
First of all, we are very grateful for this great contribution by the respected Referee and his efforts that helped us to improve our review paper. We have substantially revised our paper and highlighted all track changes. The answers are provided below.
Comment 1 : Please comment on the cost of using the proposed technologies.
Answer:
The cost of some proposed technologies is very difficult to estimate due to the relatively low technology readiness level. However, we have studied all available literature related to the techno-economic assessment for proposed (or similar) technologies. The following part was added to the Discussion, Lines 597-672.
Comment 2:
Please list effluent standard of produced water in author’s country.
The drilling operations in Arabian Gulf are controlled by ROPME protocols, that are very strict. The oil content in the effluent should be less than 15 ppm. Lines 58-63.
We provided the following paragraph with the relevant citation. All the details can be found in Ref8.
Comment 3
Please comment whether the proposed treatment technologies can meet effluent standard.
Answer
Yes, they can. The estimation is provided below. We decided to add this information to the discussion section. Lines 611-672.
Comment 4:
Please include a table to summarize treatment performance of using proposed technologies and other commonly used treatment methods reported in the literature.
Answer:
The summary table is prepared and provided in the Discussion part.
Comment 5: Please include pre-treatment needed for the proposed treatment method reviewed in this paper.
Answer:
The pre-treatment methods, required for the upscale methodology cannot be considered at present, as it would be dependent on the properties of the effluent. The pre-treatment of produced water may include de-emulsification, mechanical treatment, or chemical treatments. Anyway, nanomembranes or plasma technologies cannot be considered as a stand-alone one-step solution. The produced water treatment should be composed of a combination of several techniques and processes in order to fulfill the requirements for water quality and cost-efficiency.
We introduced a short paragraph to the discussion. Lines 663-672
Reviewer 2 Report
All notes and comments are highlighted in the attached file

Author Response
We are absolutely grateful to Referee for his great input that definitely improved the paper quality.
We have followed his suggestions and revised our text accordingly, all changes were highlighted, but there is a couple of questions/ suggestions. Please find our answers below:
This information is so obvious that it does not require literature references.
Answer
We appreciate the suggestion by the reviewer, but if possible, it would be better to keep the references, as in some cases a keen reader would be interested in such refs.
The comment “these values are lower. The 231 results showed that these membranes effectively separated the layered oil/water mixture 232 with permeate flux up to 57509 L m−2 h-1, while high fluxes of 1390–6460 L m−2 h-1 were 233 observed for the separation of different oil-in-water emulsions.
Answer
The fluxes for emollitions were lower than for the layered oil due to the more difficult separation of the emulsions than layered oil. Indeed, the emulsions generally permeate slower through the membrane working by the separation according to the differences in WCA. I hope that our explanation is acceptable and sufficient.
Round 2
Reviewer 2 Report
Line 619 - According to [68]: R(%) = (1−Cp/Cf)x 100